# Associations between Maternal Polychlorinated Dibenzo-p-dioxin and Dibenzofuran Serum Concentrations and Pulse Pressure in Early Pregnancy: A Cross-Sectional Study

**DOI:** 10.3390/ijerph192113785

**Published:** 2022-10-23

**Authors:** Xiaofang Liu, Wencheng Cao, Xiao Liu, Yan Zhou, Sheng Wen

**Affiliations:** Hubei Provincial Key Laboratory for Applied Toxicology, Hubei Provincial Center for Disease Control and Prevention, Wuhan 430079, China

**Keywords:** PCDD/Fs, pulse pressure, early pregnancy, exposure risk

## Abstract

Pulse pressure (PP) is the difference between systolic blood pressure (SBP) and diastolic blood pressure (DBP), and an independent predictor of cardiovascular risk. Previous research suggests, with different conclusions, that exposure to polychlorinated dibenzo-p-dioxins and dibenzofurans (PCDD/Fs) could affect blood pressure (BP). We conducted a cross-sectional study to determine the association of dioxin exposure with PP in early pregnancy. A total of 305 pregnant women in early pregnancy in Yingcheng, China, recruited from May 2018 to February 2021, were included in this study. We measured 17 congeners of PCDD/Fs in maternal serum via high-resolution gas chromatography tandem high-resolution mass spectrometry. A generalized linear regression model was used to analyze the influencing factors of dioxin exposure and their relationships with PP. The levels of total PCDD/Fs (∑PCDD/Fs) ranged from 163.52 pg/g lipid to 1,513,949.52 pg/g lipid, with a mean of 10,474.22 pg/g lipid. The mean toxicity equivalent (TEQ) of total PCDD/Fs (∑TEQ-PCDD/Fs) was 42.03 pg/g lipid. The ratio of tetrachlorinated to octa-chlorinated congeners in maternal serum was enriched with an increasing number of chlorines. Pregnant women with college and above education had higher concentrations of ∑PCDD/Fs than those with education levels of junior high school and below (β = 0.34, 95% CI: 0.01, 0.67). The adjusted model for ∑TEQ-PCDD/Fs was significantly and negatively associated with PP (β = −1.79, 95% CI: −2.91, −0.68). High levels of dioxins were found in this area, and exposure to dioxins may affect the PP of women in early pregnancy, with health risks.

## 1. Introduction

Current guidelines for the diagnosis and management of hypertension have defined cardiovascular risk by the elevation of systolic blood pressure (SBP) and/or the elevation of diastolic blood pressure (DBP) [1]. However, the value of pulse pressure (PP) in the clinical assessment of cardiovascular risk has become increasingly evident [2]. PP is defined as the difference between SBP and DBP, and various studies have verified PP as an independent predictor of cardiovascular risk [3]. Polychlorinated dibenzo-p-dioxins and dibenzofurans (PCDD/Fs) compounds are ubiquitous in the environment and bioaccumulate and biomagnify through the food chain due to their persistence and lipophilicity [4,5]. They were shown to induce carcinogenicity, endocrinopathy, neurotoxicity, and immunotoxicity [6]. A handful of studies suggest that exposure to dioxins may also be associated with an increased risk of chronic diseases, such as hypertension, ischemic heart disease, stroke, diabetes, and hyperlipidemia [7,8,9,10].

Pregnant women are a special group of people who may intake more food of animal origin, which contributes to more than 90% of dioxins body burden [11]. Studies undertaken on maternal participants highlighted epidemiological evidence of short- and long-term effects of in utero exposure. These include associations between maternal exposure, diet, and age with hormonal disruptions in children and impacts on estrogenic metabolism [12,13,14,15,16]. Additionally, investigative studies on fetal exposure to dioxins provide evidence of an association between maternal serum and fetal abortion, birth defects, and low birth weight [17]. Although higher concentrations of dioxins have been detected in the placenta and umbilical cord blood, venous blood samples during pregnancy are considered the most representative to evaluate maternal or fetal body burdens [18,19]. In recent years, numerous studies have supported increasing evidence of adverse effects of PCDD/Fs in pregnant women. These include the populations in Mexico [20], Japan [12,21], Vietnam [22], Taiwan [23], Beijing [24], Tianjin [6], Tema and Accra [25].

Moreover, emerging data support the plausible contribution of specific environmental toxicants to BP, including dioxins and dioxin-like compounds. A number of studies conducted in populations have shown significant associations between dioxin-like compounds and increased BP or hypertension, although the associations with individual congeners are not consistent across studies [7,9,26,27]. Animal studies show that 2,3,7,8-TCDD exposure can affect the BP of experimental animals, but the effect may be related to the dose or the species of the animals tested [28,29,30]. Gestational hypertension during pregnancy is a common obstetric disease, accounting for 5~10% of all pregnancies and causing approximately 10~16% of all pregnancy-related deaths [31]. However, no reports have been found on the relationship between PCDD/F exposure and PP during early pregnancy. In the present study, we addressed this issue by conducting a cross-sectional study to assess the risks of dioxin exposure and the association with PP in early pregnancy.

## 2. Materials and Methods

### 2.1. Study Design and Population

We conducted a birth cohort study (H-YCCS) in Yingcheng, Hubei Province, China, from May 2018 to February 2021 to investigate the associations between dioxin exposure in early pregnancy and adverse health outcomes. Healthy pregnant women were recruited, and prenatal examinations were conducted in Yingcheng People’s Hospital, China.

A total of 722 pregnant women were recruited, and 605 (83.8%) of them completed the questionnaire. The questionnaire included general demographic characteristics (age, height, weight, ethnicity, education level, monthly household income, occupation, occupational exposure to hazard factors, alcohol consumption, active and passive smoking, reproductive history, drug use, and disease). BP (SBP and DBP) measurements of pregnant women were obtained by an automatic standard sphygmomanometer beginning after the patient had been sitting for 5 min. Finally, 305 (50.4%) women provided sufficient blood samples for PCDD/F measurements. Our study was approved by the Ethics Committee of Hubei Center for Disease Control and Prevention.

### 2.2. Blood Collection and Chemical Analysis

Fasting venous blood samples were collected from the participants by nurses. The whole blood sample was centrifuged immediately, and the serum was segregated and preserved at −40 °C until further analysis. PCDD/Fs in serum were extracted and determined as previously described [6]. Briefly, approximately 2 mL of serum was mixed with diatomite and then spiked with 13C-labeled internal standards (EPA-1613LCS, Wellington Laboratories Inc., Guelph, ON, Canada). The mixture was extracted by an accelerated solvent extractor (ASE-914, BÜCHI, Switzerland) at 130 °C and 100 bar. The extract was concentrated to near one milliliter and then purified using an acid silica gel column and carbon column under a pressure of 0.1 MPa (F12, CAPE Technologies, South Portland, ME, USA). Then, the PCDD/F fraction was collected and concentrated to near dryness and redissolved in approximately 20 microliters of nonane. The 13C-labeled recovery standards for PCDD/Fs (EPA-1613ISS, Wellington Laboratories Inc., Guelph, ON, Canada) were injected before instrumental analysis. Concentrations of PCDD/Fs were analyzed by high-resolution gas chromatography tandem high-resolution mass spectrometry (HRGC-HRMS, DFS, Thermo Fisher Scientific, Waltham, MA, USA), equipped with a capillary column (DB-5 MS, 60 m × 0.25 mm ID × 0.25 μm, Agilent Technologies, Santa Clara, CA, USA). The total lipids of each maternal serum sample were detected by an enzymatic summation method [32]. All data were reported on a lipid basis.

### 2.3. Quality Control

One procedure blank test was carried out for every eight serum samples to examine contamination of the analysis system, and one test of standard reference material (Organic Contaminants in Non-Fortified Human Serum, SRM-1957) purchased from the U.S. National Institute of Standards and Technology. Procedure blank and SRM-1957 were analyzed to identify potential system contamination and as a quality control to assess laboratory precision during the entire analytical process. The precision and accuracy of measurements for PCDD/Fs in SRM-1957 were within the certified reference ranges recommended by the manufacturer. The method detection limits (MDLs) were 0.05–0.16 pg/g lipid. Recoveries of internal standards ranged from 47% to 99%, thus meeting the requirements of U.S. Environmental Protection Agency (EPA) method 1613.

### 2.4. Statistical Analysis

The values below the detection limit were assigned as half of the MDLs for statistical calculations. Toxicity equivalents (TEQs) were calculated using the World Health Organization (WHO) 2005 toxicity equivalence factors (TEFs) [33]. All statistical analyses were performed using Empower stats software (Available online: www.empowerstats.com (accessed on 17 March 2021), X&Y Solutions, Inc., Boston, MA, USA) and R software, version 3.2.0 (Available online: http://www.R-project.org/ (accessed on 17 March 2021)). Since the value of PCDD/Fs did not conform to a normal distribution, natural logarithm transformation (ln) was carried out. PCDDs (7 species, including 2,3,7,8-TCDD, 1,2,3,7,8-PeCDD, 1,2,3,4,7,8-HxCDD, 1,2,3,6,7,8-HxCDD, 1,2,3,7,8,9-HxCDD, 1,2,3,4,6,7,8-HpCDD, and OCDD), PCDFs (10 species, including 2,3,7,8-TCDF, 1,2,3,7,8-PeCDF, 2,3,4,7,8-PeCDF, 1,2,3,4,7,8-HxCDF, 1,2,3,6,7,8-HxCDF, 1,2,3,7,8,9-HxCDF, 2,3,4,6,7,8-HxCDF, 1,2,3,4,6,7,8-HpCDF, 1,2,3,4,7,8,9-HpCDF, and OCDF) and PCDD/Fs (7 PCDDs and 10 PCDFs) concentrations and TEQs were included in the analysis. ∑PCDDs and ∑TEQ-PCDDs are the sum of 7 PCDD concentrations and TEQs, respectively, ∑PCDFs and ∑TEQ-PCDFs are the sum of 10 PCDF concentrations and TEQs, and ∑PCDD/FS and ∑TEQ-PCDD/FS are the sum of 17 concentrations and TEQs. We considered multiple covariates and potential confounders for the association of PCDD/F exposure with influencing factors and BP, and factors with *p* < 0.1 were included in the final multiple linear regression models. The following variables were considered for use: maternal age (years), body mass index (BMI, kg/m^2^), days of pregnancy, education level (junior high school, high school, college and above), average monthly family income (<5000 yuan, ≥5000 yuan), parity (0, ≥1), alcohol consumption (No, Yes), active smoking (No, Yes), passive smoking (No, Yes), occupation (worker, farmer, others), and nationality (Han nationality, others). Values were considered statistically significant when *p* ≤ 0.05.

## 3. Results

### 3.1. Demographic Characteristics of the Study Participants

The demographic characteristics of the study participants are shown in Table 1. The mean with standard deviations (SD) of maternal age, pregnancy BMI, and days of pregnancy were 27.98 ± 3.55 years old, 21.55 ± 3.29 kg/m^2^, and 84.12 ± 8.71 days, respectively. More than half of the mothers (54.8%) were having their first pregnancy. The majority of the mothers (56.7%) had a high school education. Nearly one-third (36.4%) of the mothers did not work during pregnancy. Few mothers were active smokers (2.0%) or consumed alcohol (16.4%), but 43.3% of mothers passively smoked more than 15 min per day.

### 3.2. PCDD/Fs Concentrations in Maternal Serum

The concentrations and detection rates of PCDD/Fs in maternal serum are shown in Table 2. The detection rates ranged from 32% (2,3,7,8-TCDD) to 100% (OCDD), ∑PCDD/F concentrations in maternal serum ranged from 163.52 pg/g lipid to 1,513,949.52 pg/g lipid, with a mean of 10,474.22 pg/g lipid, and OCDD (10,111.19 pg/g lipid) was the highest concentration. ∑PCDD concentrations (10,321.86 pg/g lipid) were significantly higher than ∑PCDFs (152.36 pg/g lipid). The mean ∑TEQ-PCDD/Fs was 42.3 pg/g lipid, and the ∑PCDDs were two-fold higher than the ∑PCDFs.

### 3.3. PCDD/Fs Congeners in Maternal Serum

Level and TEQ contributions of congeners to total PCDD/Fs are shown in Figure 1. Among the PCDD/F levels, OCDD was the predominant congener, accounting for 96.5%. The ratios of PCDDs and PCDFs to total PCDD/Fs were 98.6% and 1.4%, respectively. The ratios of tetra- (0.3%), penta- (0.4%), hexa- (0.9%), hepta- (1.7%), and octa-chlorinated (96.6%) congeners were enriched with increasing amounts of chlorine. For TEQs, 1,2,3,7,8-PeCDD was the dominant congener, accounting for 41.3%, followed by 2,3,4,7,8-PeCDF (11.8%). The mean contribution of PCDDs to ∑TEQs was 66.8%, and that of PCDFs was 33.2%.

### 3.4. Other Factors

Influencing factors on PCDD/F concentrations and TEQs [β, 95% confidence interval (CI)] are presented in Table 3. Pregnant women with college and above education had higher concentrations of ∑PCDD/Fs than those with junior high school and below (β = 0.34, 95% CI: 0.01, 0.67). No associations were observed between the ∑TEQ-PCDDs, ∑TEQ-PCDFs, ∑TEQ-PCDD/Fs, and influencing factors.

### 3.5. Relationships between PCDD/Fs in Maternal Serum and BP

The relationships between BP and the TEQs in maternal serum are shown in Table 4. ∑TEQ-PCDD/Fs in maternal serum was significantly and negatively correlated with PP in both crude and adjusted models (β = −1.62, 95% CI: −2.72, −0.52 and β = −1.79, 95% CI: −2.91, −0.68, respectively) and the same as ∑TEQ-PCDDs (β = −1.10, 95% CI: −1.94, −0.26, and β = −1.23, 95% CI: −2.08, −0.37, respectively). A significantly negative association between ∑TEQ-PCDFs and PP was found only in the adjusted model (β = −1.20, 95% CI: −2.39, −0.01). ∑TEQ-PCDFs were also found to be significantly and negatively correlated with SBP in both crude and adjusted models (β = −1.24, 95% CI: −2.40, −0.09 and β = −1.31, 95% CI: −2.47, −0.15, respectively). However, exposure to ∑TEQ-PCDDs was significantly and positively associated with DBP after adjustment (β = 0.79, 95% CI: 0.04, 1.54). No associations were observed between the concentrations of ∑PCDDs, ∑PCDFs, ∑PCDD/Fs, and BP.

## 4. Discussion

We measured the PCDD, PCDF, PCDD/F, and TEQ concentrations in maternal serum early in pregnancy for a Chinese birth cohort study (H-YCCS). The total TEQs of PCDDs, PCDFs, and PCDD/Fs in maternal serum between this study and other studies globally are shown in Table 5. Mean ∑TEQ-PCDDs, ∑TEQ-PCDFs, and ∑TEQ-PCDD/Fs in the present study were higher than most background concentrations reported in the serum of pregnant women in other locations of the world. ∑TEQ-PCDD/Fs in maternal serum were also compared with the biomonitoring equivalent (BE) for risk assessment. BE values are designed to be used as screening tools to assess whether chemicals have a large, small, or no margin of safety compared to existing health-based exposure guidelines [34]. On the basis of neurodevelopmental effects, ∑TEQs of 15 pg/g lipid in serum samples is consistent with the minimal risk level recommended by the Agency for Toxic Substances and Disease Registry [35]. Referring to the ∑TEQ-PCDD/Fs in our data, 175 (57.4%) of the participants showed higher levels than the BEs. Thus, future studies should address long-term exposure to dioxins in pregnant women.

Concentrations of tetrachlorinated to octa-chlorinated congeners in serum enriched with increasing amounts of chlorine. These accumulations of highly chlorinated congeners in serum might be attributed to their high molecular weight [18,36,37]. It has been reported that once higher chlorinated congeners enter serum, they are more likely to accumulate with time [38]. Both the number and position of chlorine substituted on the ring of PCDD/Fs was important for metabolism [39]. We found that pregnant women with a higher education had higher concentrations of ∑PCDD/Fs. Educational status may be a marker for socioeconomic status in this population, and both education and socioeconomic status have been positively associated with maternal dietary quality [40]. They have better conditions and tend to increase the consumption of animal-origin food [41]. It has been estimated that more than 90% of current human exposure to dioxins among the general population occurs via food consumption, primarily from animal-origin food [42]. PCDD/Fs are highly lipophilic, environmentally persistent, and more likely to be enriched in adipose tissues [43].

Only a few studies have reported the relationship between PCDD/Fs and BP. Vietnamese soldiers exposed to high levels of 2,3,7,8-TCDD may have an increased risk of hypertension but not other PCDD/F congeners [26]. It has been reported that high TEQs of PCDD/Fs in Japanese general populations may increase the risk of hypertension [7]. However, the relationship between dioxin exposure and PP during pregnancy has not been reported until now. In this study, high exposure to PCDD/Fs was associated with decreased PP in pregnant women. Our results differ from the conclusions reported by Cypel and Nakamoto. Animal studies have shown that dioxins could affect BP. Sustained aryl hydrocarbon receptor (AhR) activation by 2,3,7,8-TCDD exposure induces hypertension in adult male C57BL/6 mice. This model provides valuable insight into the mechanisms underlying 2,3,7,8-TCDD-induced cardiovascular pathogenesis, including the role of vascular reactive oxygen species (ROS) as potential mediators of 2,3,7,8-TCDDMinduced hypertension [29]. When C57BL/6J mice are continuously exposed to 2,3,7,8-TCDD, AHR regulates the expression of several genes, including CYP1A1 and cyclooxygenase (COX-2), which may influence BP through the production of vasoactive eicosanoids [30]. However, different results showed that BP in the infected group was lower than that in the control group after exposure to 2,3,7,8-TCDD in female rats [28]. These studies suggest that 2,3,7,8-TCDD exposure could affect the BP of experimental animals, but the effect may be related to the dose or the species of the tested animals.

Mechanisms that might be involved in the associations between PCDD/Fs and PP are currently largely unknown. Experimental evidence suggests that dioxin-like compounds can modulate endothelium-derived vasoactive factors in human primary endothelial cells and induce changes characteristic of endothelial dysfunction in human essential hypertension [44]. It can modulate the expression of vasoconstriction factors such as COX-2, prostaglandins, and reactive oxygen species (ROS) and change the production of nitric oxide (NO), a well-known vasodilator factor [45]. It has been reported that a decreased PP may be associated with hypovolemia, cardiac failure, cardiac arrhythmia, valvular heart disease, aortic dissection, or low BP [46]. Upon review of previous medical literature, a low PP has been shown to be an indicator of decreased cardiac function and poor outcomes in patients with myocardiac infarction and a predictor of cardiovascular death in patients with mild to advanced heart failure [47]. Although low PP was less apparent in pregnant women, if these associations are indeed present at relatively common serum PCDD/Fs concentrations, the public health significance of the relationships may be considerable. Clearly, it is very important that these possibilities should be tested systematically in a prospective investigation. There is also a need for investigations that can identify the mechanisms that might underlie these associations. For PP, a number of external factors can influence levels or performance, but the observation that serum PCDD/Fs levels can influence them indicates that PCDD/Fs levels may cause physiological changes and contribute to the development of diseases even when they are not the sole cause.

This study has several limitations. First and foremost is the limited sample size of our study population. As the serum dioxin concentration is at the pg/g level, sufficient serum is required. Only a subset of the population provided sufficient samples. Second, the current study on the relationship between dioxin exposure and PP is cross-sectional, which cannot prove causality regardless of how strong the associations are.

## 5. Conclusions

Our collective findings reveal that pregnant women may be at risk of exposure to PCDD/Fs. The ratio of tetrachlorinated to octa-chlorinated congeners in maternal serum was enriched with an increasing number of chlorines. Pregnant women with higher education had higher levels of ∑PCDD/Fs. We found that ∑TEQs were significantly and negatively associated with PP. These exploratory results require corroboration with further studies involving larger sample sizes and a wider range of exposure to establish the association between exposure to dioxins and maternal as well as offspring health.

## Figures and Tables

**Figure 1 ijerph-19-13785-f001:**
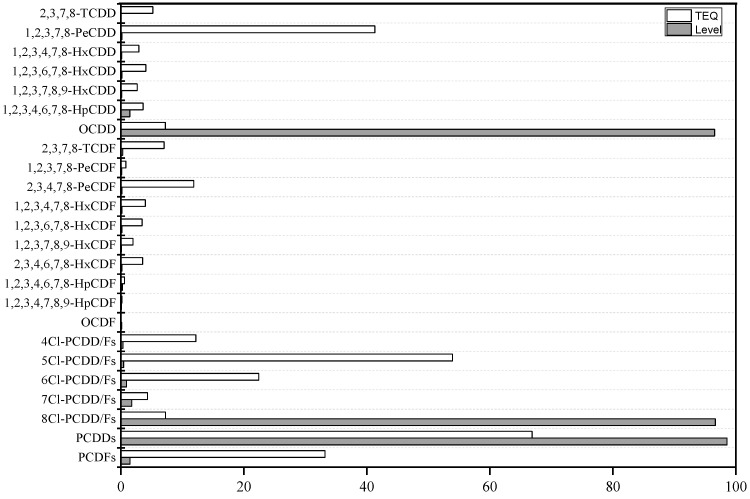
The contributions of congeners to total PCDD/Fs in maternal serum (%).

**Table 1 ijerph-19-13785-t001:** Demographic characteristics among the study participants (*n* = 305).

Variable	Statistics	PP	SBP	DBP
Nationality				
Han nationality	302 (99.0%)	47.81 ± 10.36	120.99 ± 10.22 *	73.18 ± 9.13
Others	3 (1.0%)	53.33 ± 5.51	134.33 ± 4.51	81.00 ± 7.21
Maternal education				
Junior high school and blow	132 (43.3%)	47.95 ± 9.94	122.22 ± 9.72	74.27 ± 8.81
High school	83 (27.2%)	47.02 ± 10.99	119.93 ± 9.58	72.90 ± 9.77
College and above	90 (29.5%)	48.50 ± 10.34	120.61 ± 11.53	72.11 ± 8.94
Household income, RMB, per month				
≤5000 Yuan	155 (50.8%)	47.33 ± 9.89	120.62 ± 8.92	73.29 ± 9.48
>5000 Yuan	150 (49.2%)	48.41 ± 10.78	121.64 ± 11.50	73.23 ± 8.80
Parity				
0	167 (54.8%)	47.68 ± 9.98	121.33 ± 10.02	73.65 ± 9.20
≥1	138 (45.2%)	48.09 ± 10.77	120.87 ± 10.59	72.78 ± 9.06
Occupation				
Worker	152 (49.8%)	47.25 ± 9.91	119.70 ± 9.38	72.45 ± 9.15 *
Farmer	111 (36.4%)	47.53 ± 9.89	122.52 ± 10.44	74.99 ± 9.11
Others	42 (13.8%)	50.95 ± 12.51	122.55 ± 12.23	71.60 ± 8.61
Active smoking				
No	299 (98.0%)	47.88 ± 10.36	121.29 ± 10.20 *	73.41 ± 9.01 *
Yes	6 (2.0%)	47.00 ± 9.76	112.83 ± 11.30	65.83 ± 13.00
Passive smoking				
No	173 (56.7%)	47.76 ± 10.71	121.58 ± 10.45	73.82 ± 9.14
Yes	132 (43.3%)	48.00 ± 9.86	120.52 ± 10.03	72.52 ± 9.11
Alcohol Drinking				
No	255 (83.6%)	48.03 ± 10.31	121.51 ± 10.39	73.48 ± 9.32
Yes	50 (16.4%)	47.00 ± 10.54	119.14 ± 9.43	72.14 ± 8.15
Marital status				
No	2 (0.7%)	53.00 ± 5.66	122.50 ± 12.02	69.50 ± 17.68
Yes	303 (99.3%)	47.83 ± 10.35	121.11 ± 10.28	73.28 ± 9.10
Previous history				
No	282 (92.5%)	47.72 ± 10.38	121.16 ± 10.43	73.43 ± 9.24
Yes	23 (7.5%)	49.57 ± 9.84	120.70 ± 8.20	71.13 ± 7.65
Hazard factors				
No	283 (92.8%)	47.93 ± 10.50	73.29 ± 9.20	121.22 ± 10.45
Yes	22 (7.2%)	47.00 ± 8.02	72.86 ± 8.51	119.86 ± 7.64
Maternal age (years)	27.98 ± 3.55	0.16 (−0.16, 0.49)	0.02 (−0.31, 0.34)	−0.15 (−0.44, 0.14)
BMI (kg/m^2^)	21.55 ± 3.29	0.28 (−0.08, 0.63)	0.11 (−0.24, 0.46)	−0.17 (−0.48, 0.14)
Days of pregnancy	84.12 ± 8.71	0.08 (−0.05, 0.21)	−0.02 (−0.15, 0.11)	−0.10 (−0.22, 0.01)

* *p* < 0.05.

**Table 2 ijerph-19-13785-t002:** Concentrations of PCDD/Fs in maternal serum, expressed as pg/g lipid (*n* = 305).

Compounds	Mean	Minimum	Median	Maximum	DR (%)
2,3,7,8-TCDD	2.18	0.05	0.05	80.16	32%
1,2,3,7,8-PeCDD	17.35	0.07	3.96	695.45	61%
1,2,3,4,7,8-HxCDD	12.18	0.02	2.12	320.22	68%
1,2,3,6,7,8-HxCDD	16.94	0.02	4.13	374.10	83%
1,2,3,7,8,9-HxCDD	11.08	0.02	1.94	342.81	73%
1,2,3,4,6,7,8-HpCDD	150.94	0.05	58.00	12,557.35	99%
OCDD	10,111.19	79.40	1273.92	1,501,331.77	100%
∑PCDDs	10,321.86	112.84	1405.47	1,513,889.31	
2,3,7,8-TCDF	29.44	0.04	22.10	253.74	80%
1,2,3,7,8-PeCDF	10.99	0.08	4.15	151.57	70%
2,3,4,7,8-PeCDF	16.57	0.07	7.47	247.00	85%
1,2,3,4,7,8-HxCDF	16.66	0.05	7.03	316.09	92%
1,2,3,6,7,8-HxCDF	14.42	0.04	4.18	334.58	87%
1,2,3,7,8,9-HxCDF	8.11	0.03	2.02	277.86	76%
2,3,4,6,7,8-HxCDF	14.72	0.04	2.55	371.83	83%
1,2,3,4,6,7,8-HpCDF	23.84	0.04	11.52	548.31	97%
1,2,3,4,7,8,9-HpCDF	6.22	0.05	1.40	391.51	69%
OCDF	11.40	0.03	6.05	679.44	86%
∑PCDFs	152.36	9.17	82.79	3279.23	
tetra-chlorinated PCDD/Fs	31.62	0.09	24.38	257.44	
penta-chlorinated PCDD/Fs	44.91	0.22	17.28	746.47	
hexa-chlorinated PCDD/Fs	94.11	3.48	26.24	2165.06	
hepta-chlorinated PCDD/Fs	180.99	0.14	80.99	12,576.23	
octa-chlorinated PCDD/Fs	10,122.59	84.81	1297.16	1,501,331.80	
∑PCDD/Fs	10,474.22	163.52	1632.38	1,513,949.52	
∑TEQ-PCDDs	28.09	0.35	9.04	767.3	
∑TEQ-PCDFs	13.94	0.60	7.48	214.39	
∑TEQ-PCDD/Fs	42.03	2.02	18.17	786.11	

DR, detection rate; TEQ: Toxicity equivalent; PCDD/Fs: Polychlorinated Dibenzo-p-Dioxins and dibenzofurans; TCDD: Tetrachlorodibenzo-p-dioxin; PeCDD: Pentachlorodibenzo-p-dioxin; HxCDD: Hexachlorodibenzo-p-dioxin; HpCDD: Heptachlorodibenzo-p-dioxin; OCDD: Octachlorodibenzodioxin; TCDF: Tetrachlorodibenzofuran; PeCDF: Hexachlorodibenzofuran; HxCDF: Hexachlorodibenzofuran; HpCDF: Heptachlorodibenzofuran; OCDF: Octachlorodibenzofuran; tetra-chlorinated PCDD/Fs: 2,3,7,8-TCDD + 2,3,78-TCDF; penta-chlorinated PCDD/Fs: 1,2,3,7,8-PeCDD + 1,2,3,7,8-PeCDF + 2,3,4,7,8-PeCDF; hexa-chlorinated PCDD/Fs: 1,2,3,4,7,8-HxCDD + 1,2,3,6,7,8-HxCDD + 1,2,3,4,7,8-HxCDF + 1,2,3,6,7,8-HxCDF + 1,2,3,7,8,9-HxCDF + 2,3,4,6,7,8-HxCDF; hepta-chlorinated PCDD/Fs: 1,2,3,4,6,7,8-HpCDD + 1,2,3,4,6,7,8-HpCDF + 1,2,3,4,7,8,9-HpCDF; octa-chlorinated PCDD/Fs: OCDD + OCDF.

**Table 3 ijerph-19-13785-t003:** Results of the multivariate linear regression model for the groups of ln-transformed concentrations in maternal serum (*n* = 305).

Variable	∑PCDDs	∑PCDFs	∑PCDD/Fs
Maternal age (years)	0.02 (−0.00, 0.03)	−0.01 (−0.02, 0.01)	0.01 (−0.00, 0.03)
Days of pregnancy	0.03 (−0.01, 0.08)	0.02 (−0.00, 0.05)	0.03 (−0.00, 0.07)
Maternal education			
Junior high school and blow	1.00	1.00	1.00
High school	−0.07 (−0.44, 0.29)	0.14 (−0.10, 0.37)	−0.06 (−0.40, 0.28)
College and above	0.35 (−0.00, 0.71)	0.18 (−0.05, 0.41)	0.34 (0.01, 0.67) *
Passive smoking			
No	1.00	1.00	1.00
Yes	−0.09 (−0.39, 0.21)	0.14 (−0.05, 0.34)	−0.08 (−0.36, 0.21)

* *p* < 0.05.

**Table 4 ijerph-19-13785-t004:** Regression coefficients [β (95% CI)] for BP associated with groups of ln-transformed TEQs in maternal serum (*n* = 305).

Analytes	Nonadjusted	Adjusted ^a^
PP		
∑TEQ-PCDDs	−1.10 (−1.94, −0.26) *	−1.23 (−2.08, −0.37) **
∑TEQ-PCDFs	−1.00 (−2.17, 0.16)	−1.20 (−2.39, −0.01) *
∑TEQ-PCDD/Fs	−1.62 (−2.72, −0.52) **	−1.79 (−2.91, −0.68) **
SBP		
∑TEQ-PCDDs	−0.36 (−1.20, 0.48)	−0.44 (−1.28, 0.41)
∑TEQ-PCDFs	−1.24 (−2.40, −0.09) *	−1.31 (−2.47, −0.15) *
∑TEQ-PCDD/Fs	−0.86 (−1.97, 0.24)	−0.91 (−2.01, 0.20)
DBP		
∑TEQ-PCDDs	0.74 (−0.01, 1.49)	0.79 (0.04, 1.54) *
∑TEQ-PCDFs	−0.24 (−1.27, 0.79)	−0.11 (−1.16, 0.94)
∑TEQ-PCDD/Fs	0.76 (−0.23, 1.74)	0.89 (−0.10, 1.87)

^a^ Adjusted by age, parity, drinking, smoking, occupation, income, education, BMI, days of pregnancy, and nationality; * *p* < 0.05; ** *p* < 0.01.

**Table 5 ijerph-19-13785-t005:** A comparison of background TEQs with relevant studies.

Location	Sampling Year	Sample Size	∑TEQs (pg/g Lipid)			Reference
			PCDDs	PCDFs	PCDD/Fs	
Tianjin	2017	24	7.34	6.62	14.0	[6]
Hokkaido	2002–2005	379	7.26	2.53	9.79	[12]
Mexico	2005–2006	240	5.0	1.3	6.3	[20]
Sapporo	2002–2005	119	8.2	2.9	11.1	[21]
Vietnam	2012	16	10.05	4.06	14.5	[22]
Beijing	2013–2015	55	4.1	5.7	9.8	[24]
Tema/Accra	2017	34	2.10	0.99	3.09	[25]
Yingcheng	2018–2020	305	28.09	13.94	42.03	This Study

## Data Availability

Data are available upon request contacting zy_hbcdc@163.com (Y.Z.).

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
