# Peer review of "Associations between Maternal Polychlorinated Dibenzo-p-dioxin and Dibenzofuran Serum Concentrations and Pulse Pressure in Early Pregnancy: A Cross-Sectional Study"

_ijerph, 2022, doi:10.3390/ijerph192113785_

Round 1

Reviewer 1 Report

Line 11 (in the Abstract)

Previous research suggests that exposure to polychlorinated dibenzo-p-dioxins and dibenzofurans (PCDD/Fs) could affect blood pressure (BP), with different conclusions. 

Suggested Rewording: 

Previous research suggests, with different conclusions, that exposure to polychlorinated dibenzo-p-dioxins and dibenzofurans (PCDD/Fs) could affect blood pressure (BP).

Reviewer 2 Report

The authors present novel findings in a cross-sectional cohort of pregnant women with exposure to (high) background levels of dioxins/furans. The paper is well written.

1. The authors found a negative correlation between blood pressure and dioxin/furan exposure. The introduction and discussion are more in line with a positive than a negative correlation. Are the findings unexpected? If so, the authors might want to elaborate on this and offer an explanation or hypothesis. More elaboration on the negative correlation could improve the manuscript.

2. The authors discuss the negative health effects of hypertension. However, a negative association with dioxins/furans was found. Do the authors expect (negative) health effects in the range of the blood pressure measurements seen? What do the outcomes suggest?

3. The authors report high to extremely high concentrations of dioxins/furans in the mothers of the cohort. How do the authors explain this? Would the authors expect other blood pressure outcomes with other background levels (a U-curve)? Elaboration in the manuscript would make the paper more sound, I would expect.

Reviewer 3 Report

The article is well organized, references are well chosen, methodology is clearly explained and appropriate. The article is well written and easy to understand. However, I noticed some gaps.

1.      ABSTRACT

Line 14:  I didn’t find the abbreviation PP explained in the earlier lines.

Line 16: I suggest replacing the word "Types" with the word “congeners”

2. INTRODUCTION

Line 36-39 I would suggest a different citation, e.g.:

·        Schecter A, Birnbaum L, Ryan JJ, Constable JD. Dioxins: an overview. Environ Res. 2006 Jul;101(3):419-28. doi: 10.1016/j.envres.2005.12.003. Epub 2006 Jan 30. PMID: 16445906.

·        EFSA CONTAM Panel (EFSA Panel on Contaminants in the Food Chain), Knutsen, HK, Alexander, J, Barregård, L, Bignami, M, Brüschweiler, B, Ceccatelli, S, Cottrill, B, Dinovi, M, Edler, L, Grasl-Kraupp, B, Hogstrand, C, Nebbia, CS, Oswald, IP, Petersen, A, Rose, M, Roudot, A-C, Schwerdtle, T, Vleminckx, C, Vollmer, G, Wallace, H, Fürst, P, Håkansson, H, Halldorsson, T, Lundebye, A-K, Pohjanvirta, R, Rylander, L, Smith, A, van Loveren, H, Waalkens-Berendsen, I, Zeilmaker, M, Binaglia, M, Gómez Ruiz, JÁ, Horváth, Z, Christoph, E, Ciccolallo, L, Ramos Bordajandi, L, Steinkellner, H and Hoogenboom, LR, 2018. Scientific Opinion on the risk for animal and human health related to the presence of dioxins and dioxin-like PCBs in feed and food. EFSA Journal 2018;16(11):5333, 331 pp. https://doi.org/10.2903/j.efsa.2018.5333

Materials and methods

Line 72: The authors here give a broader time frame, other than in the abstract.

Results:

Line 226-229: Missing information for me is whether women with higher education were pregnant for the first time.

FIGURES and TABLES

I have no comments on the table and figure
